

# Ground-based Observations of Cloud and Drizzle Liquid Water Path in Stratocumulus Clouds

Maria P. Cadeddu [1], Virendra P. Ghate[1], Mario Mech[2]

5 [1]Environmental Sciences Division, Argonne National Laboratory, Argonne, IL, 60439, USA
[2]University of Cologne, Cologne, 50969, Germany

*Correspondence to*: Maria P. Cadeddu (mcadeddu@anl.gov)

**Abstract.** The partition of cloud and drizzle liquid water path in precipitating clouds plays a key role in determining the cloud lifetime and its evolution. A technique to quantify cloud and drizzle liquid water path by combining measurements 10 from a three-channel microwave radiometer (23.8, 30, and 90 GHz) with those from a vertically pointing Doppler cloud radar and a ceilometer is presented. The technique is showcased using one-day of observations to derive precipitable water vapor, liquid water path, cloud water path, drizzle water path below the cloud base, and drizzle water path above the cloud base in precipitating stratocumulus clouds. The resulting cloud and drizzle water path within the cloud are in good qualitative agreement with the information extracted from the radar Doppler spectra. The technique is then applied to ten 15 days each of precipitating closed and open cellular marine stratocumuli. In the closed cell systems only ~20% of the available drizzle in the cloud falls below the cloud base, compared to ~40% in the open cell systems. In closed cell systems precipitation is associated with radiative cooling at the cloud top < -100 W/m$^2$ and liquid water path > 200 g/m$^2$. However, drizzle in the cloud begins to exists at weak radiative cooling and liquid water path > ~150 g/m$^2$. Our results collectively demonstrate that neglecting scattering effects for frequencies at and above 90 GHz leads to overestimation of the total liquid 20 water path of about 10-15%, while their inclusion paves the path for retrieving drizzle properties within the cloud.

## 1 Introduction

Marine stratocumulus clouds have significant impact on the Earth's radiation balance as they reflect greater amount of solar radiation back to space compared to the ocean surface, and emit similar amount of longwave radiation as the surface. The processes affecting their highly organized spatial structure, and their spatial and temporal variability are a topic of active 25 research (Wood et al. 2015). Precipitation is hypothesized to play an important role in the transition between different mesoscale organizations of boundary layer clouds (Feingold and A. McComiskey 2016; Wang and Feingold, 2009). Similarly, precipitation, together with entrainment, impact the cloud microphysical properties that determine the cloud radiative effects (Wood, 2012; Yamaguchi et al., 2017). Hence, characterizing the properties of drizzling stratocumulus clouds through observations and high-resolution models for furthering our understanding of the precipitation processes has 30 been a focus of several previous studies (e.g. Ahlgrimm and Forbes, 2014; Zheng et al. 2017). From the point of view of ground-based instrumentation the study of microphysical and macro-physical cloud properties, involves combining data



from multiple instruments to retrieve information on the moments of the drop size distribution (DSD). For this purpose, new algorithms are developed that can extract key cloud and drizzle properties such as liquid water content and drop effective radius from a combination of active (e.g. radar, lidar), and passive (broadband or narrowband radiometers) sensors (e.g. Frisch et al., 1995; Fielding et al., 2014). Microwave radiometers have been extensively used in the past in such retrieval techniques to obtain the total column (i.e. cloud and drizzle) liquid water path of a precipitating cloud. By adding a 90 GHz or 183 GHz channel to the traditional 23 and 30 GHz channels, the uncertainty in the retrieved LWP (and column water vapor) can been reduced significantly (Löhnert and Crewell, 2003). However, recent theoretical studies (Cadeddu et al., 2017) have shown that drizzle sized hydrometeors (larger than 90 microns in diameter) significantly scatter the radiation at 90 GHz and could also be used to derive separate estimates of integrated drizzle water and cloud water.

In this work we propose a technique to retrieve column integrated values of i) drizzle water below the cloud base, ii) drizzle water above the cloud base, and iii) cloud water above the cloud base by combining the data from vertically pointing cloud radar, lidar, and a microwave radiometer. The technique is applied to 20 days of data collected at the Atmospheric Radiation Measurement (ARM) Eastern North Atlantic (ENA) site during light to moderate precipitating stratocumulus cloud conditions. In **section 2** an overview of the methodology is provided followed by application to one day of data. I**n Section 3** the results are qualitatively assessed by comparison with radar-observed Doppler spectra. The entire dataset of 20 days is examined in **section 4** through averages of in-cloud and below-cloud-base drizzle properties for the precipitating shafts, and the relation between LWP, turbulence, and drizzle production is shown. The results are summarized and briefly discussed in **Section 5**.

## 2 Methodology

In Sect. 2.1 an overview of the instrumentation and the radiative transfer models is provided. The use of active sensors to derive microphysical properties of drizzle below cloud base is well established and is used in the first part of the algorithm, the active module, described in Sect. 2.2. In the second part of the algorithm, named the passive module, resides the novel approach of using scattering properties of drizzle drops to separate cloud and drizzle water path within the cloud. The passive module is described in Sect. 2.3.

### 2.1 Instrumentation and Radiative Transfer Models

The ARM ENA site has been operational since summer of 2015 and is located at the northern tip of the northernmost island Graciosa (39° N, 28° W, 15 m) in the Azores. The site has many instruments and here we describe those used in this work. A vertically pointing Ka-band Doppler radar named Ka-band ARM Zenith Radar (KAZR) continuously records the raw Doppler spectrum and its first three moments at a 2 s temporal and 20 m range resolution. Collocated to the KAZR is a laser ceilometer (lidar) that operates at 905 nm wavelength and reports the first three optical cloud base heights and the raw


backscatter at 15 s temporal and 30 m range resolution. A three-channel microwave radiometer is also present at the site that records the calibrated brightness temperatures at 23.8, 30 and 90 GHz frequencies at 10 s temporal resolution. Balloon borne

radiosondes are launched at the site every 12 hours at 00 and 12 UTC. Due to the sparseness of the radiosonde launches, the radiosonde data is interpolated with that from the ECMWF model to deduce profiles of temperature, pressure, humidity and winds at a uniform 1-minute temporal and 50 m vertical resolution. The visible imagery and cloud top temperature reported by the Spinning Enhanced Visible Satellite Imager (SEVIRI) onboard geostationary Meteosat satellite were used to confirm the presence of similar cloud conditions around the site as those observed at the site.

The ceilometer backscatter was filtered for noise using the technique proposed by Kotthaus et al. (2016), and was calibrated following O'Connor et al., (2005) using data collected on 7 March 2016. More details about the ceilometer calibration are mentioned in the Appendix of Ghate and Cadeddu (2019), referred to as GC19 from hereon. The KAZR was calibrated by comparing its reflectivity with that from the Ka-band Scanning ARM Cloud Radar that was calibrated using a corner reflector. The KAZR calibration hence is good within 1 dB. The KAZR and ceilometer data were combined to

produce estimates of the first three moments of Doppler spectra and of ceilometer backscatter on a uniform 1 min temporal and 50 m range resolution following Clothiaux et al. (2000). These were further used to calculate cloud boundaries. Microwave radiometer data are collected by a 3-channel radiometer (23.8, 30, 90 GHz). The radiometer is calibrated using tip curves (Han and Westwater, 2000) resulting in a calibrated brightness temperature uncertainty of about 0.3 K in the K-band and 1 K in the W-band. The resulting uncertainty in the derived products is about 0.4 kg/m$^2$ for PWV and 15 g/m$^2$ for

LWP. Precipitable water vapor and liquid water path derived using a neural network algorithm (Cadeddu et al., 2009) are provided in the data file. These retrievals are used as a priori information in the algorithm described in this work. For the evaluation of the scattering effect, however the MWRRETv2 (Turner, 2007) is used. This latest product uses a convergence process similar to the one used here, but with an absorption-only radiative transfer model, MonoRTM (Clough et al., 2005).

We use the *Passive and Active Microwave TRAnsfer (PAMTRA) Package* (Mech et al., 2018) available at

https://github.com/igmk/pamtra, a scattering microwave radiative transfer model that simulates active and passive measurements in plane parallel geometry between 1 and 800 GHz. The calculations are based on the fully polarized model of Evans and Stephens (1995) for non-spherical and oriented particles. The model simulates passive measurements in upward and downward geometry at a given height and allows the choice between different assumptions and models in the calculations of surface emissivity, ice crystal habit, size distribution, and calculation of scattering properties. The Rapid

Radiative Transfer Model (RRTM) (Iacono et al., 2000) was used to calculate the radiative fluxes and heating rates. We refer the reader to GC19 regarding the details of the setup and inputs of RRTM.

An example of the noise filtered profiles of KAZR reported reflectivity, ceilometer reported backscatter and the concurrent retrievals of LWP in non-scattering approximation are shown in Fig. 1. Moderate to heavily precipitating stratocumulus clouds were observed throughout the day, with some of the precipitation evaporating before reaching the

surface.



## 2.2 The active module

The active module of the retrieval technique is similar to that proposed by O'Connor et al. (2005) and applied to the ARM data by GC19 with some subtle differences. Drizzle below the cloud base is assumed to have a three-parameter gamma drop size distribution. The ceilometer backscatter, radar reflectivity, mean Doppler velocity and width of the Doppler spectra were used in an iterative manner to retrieve the three parameters of the gamma distribution. Details of the radar-lidar microphysical retrievals of drizzle properties below the cloud base are given in GC19 together with an extensive discussion of the range of validity of the algorithm. The lidar signal attenuates at the cloud base as the lidar ratio (extinction to backscatter) of cloud drops is 50-60 Sr compared to ~20 Sr of drizzle drops at the 905 nm wavelength. Hence, the ceilometer backscatter peaks at the cloud base due to the presence of cloud drops in addition to the drizzle drops. The returns at the cloud base from pixels containing both cloud and drizzle drops were neglected by GC19. In this work we assume the DSD of these cloud and drizzle mix to have a lognormal shape with a width of 0.38 and retrieve the modal diameter and number concentration. These serve as an a priori information in the retrieval framework.

The retrieved modal diameter and rain rate for the case shown in Fig. 1 are shown in Fig. 2 a and b. During this day, the drizzle modal diameter was between 100 and 800 µm and rain rate was around 2.5 mm/day with brief peaks greater than 10 mm/day. Precipitation shafts were identified using the criteria explained in G19 and shown as black solid lines in Fig. 2 a. In this specific case 24 shafts were identified with measurable precipitation detected at the surface for all the shafts. Although this does not constitute a problem for the active instrumentation it does affect the passive module because excessive water deposition on the radiometer can affect the data. At the cloud base the average modal diameter of the mixed drizzle-cloud DSD was 77.8 µm.

## 2.3 The passive module

The output from the active (radar-ceilometer) module is used as input to the microwave radiative transfer model. The theoretical basis for the retrieval is provided in Cadeddu et al. (2017). In this operational implementation only three quantities are retrieved: PWV, total liquid water path (LWPt), and $C_f$, the ratio of cloud to total LWP. The radiative transfer code, PAMTRA, used in the passive module requires information on the cloud and drizzle DSD, specifically liquid water content (LWC), the gamma parameter, and effective diameter. Because the microwave measurements are insensitive to the gamma parameter of the DSD this last is set to zero in the passive module denoting exponential distribution. The below cloud drizzle liquid water content ($DWC_{bc}$), below cloud liquid water path ($DWP_{bc}$) and the average drizzle effective radius below cloud base calculated from the active module are provided to the radiative transfer model. These properties of drizzle below are kept intact during the entire iterative process within the passive module. Figure 3 shows a flow chart of the active and passive modules with the quantities provided as input, the intermediate outputs and the final output. Additional details of the passive module are provided in Table 1.





Because *in-cloud* properties are not easily derived and the active module is only valid at and below cloud base several assumptions had to be made about the *in-cloud* DSD parameters. The drizzle LWC *above cloud base* (DWC$_{ac}$) is assumed constant with value equal to the drizzle LWC at the cloud base (Wood, 2005) and the cloud LWC (CWC) is assumed to follow an adiabatic profile (Zuidema et al., 2005). The initial adiabatic profile is determined by subtracting the initial drizzle water path (Table 1, row 6) from the initial total LWP (LWP$_t$ in Table 1, row 2) and distributing the resulting cloud liquid water path (CWP) adiabatically between cloud base and top. These estimates of CWP and the first guess LWP$_t$ are used to provide the first guess estimate of C$_f$ as shown in the flowchart (Table 1, row 9). At each iteration the drizzle liquid water path above cloud (*DWP$_{ac}$*) and CWP are adjusted based on *LWP$_t$* and C$_f$ to ensure consistency with the drizzle below cloud base by scaling the liquid water content accordingly. Once the retrieval converges the diagonal elements of the covariance matrix can provide information on the reduction of the uncertainty of the three retrieved parameters.

The retrieval of the C$_f$ parameter depends on how much the scattering information affects the measurement and is therefore dependent on the drop size distribution. It is expected that the retrieval will be more effective during precipitation characterized by drops larger than 100 μm in diameter. The advantage of having larger drops is however offset by the fact that they are usually associated with higher rain reaching the surface that impacts the convergence because of water deposition on the radiometer window. This limitation of the ground-based instrument is evident in Fig. 2c where the total LWP from this work is shown during precipitating shafts. On November 21, 2016 the retrieval converged in 367 out of the 484 minutes identified in the shafts. Using the proposed technique from aircraft or satellite will enable to study a wider range of precipitating conditions and to take better advantage of the scattering information. In fact, based on a similar principle, Jacob et al., (2019) applied a neural network retrieval to microwave measurements collected from aircraft to separate cloud from drizzle water path over the Atlantic Ocean.

Total, cloud, and drizzle LWP during the first 4 hours of 21 November 2016 (minute 1-240) are shown in Fig. 4a and b. Although the below-cloud drizzle is well defined in the active retrieval process, the information that can be gained from the microwave retrieval on the partition of cloud and drizzle depends on how much information is available from the measurements. The cloud LWP constitutes the largest portion of the total LWP, and the resulting total drizzle water path (in cloud and below cloud) is in this case about twice the precipitating drizzle. In the next section the in-cloud partition between drizzle and cloud water path is closely examined next to the radar Doppler spectra during 21 November 2016.

## 3 Comparison with the Radar Doppler Spectra

Due to lack of coincident other retrievals of cloud and drizzle water within the cloud layer, here we qualitatively evaluate them by separating the cloud and drizzle contributions in the Doppler spectra. Possible ways and the challenges of quantitavely evaluating these retrievals are discussed in the last section.



### 3.1 Radar spectra processing

Doppler spectra from cloud radars have been previously used to gain insight into the onset and evolution of drizzle in clouds
(Kollias et al., 2011a; Kollias et al., 2011b; Luke and Kollias, 2013; Acquistapace et al., 2019). The methodology is based on
the fact that the Doppler velocity of drizzle droplets is negative due to the terminal fall velocity of drizzle drops, while the
Doppler spectra of a non-precipitating cloud is centered on zero mean velocity due to their movement with turbulence. The
presence of drizzle drops in a cloud therefore introduces a negative skewness in the cloud Doppler spectra. In this section,
cloud Doppler spectra are analyzed with the intent of separating the cloud and drizzle components to qualitatively evaluate
their co-variability.

In the following analysis the Doppler spectra were averaged for one minute to reduce the effect of turbulence and they were
denoised using the technique of Hildebrand and Sekhon, (1974). Doppler spectra for six shafts that lasted for more than 20
min on 21 November 2016 and for which the microwave retrieval converged at least 75% of the times are analyzed. Figure
5 shows examples of Doppler spectra from the shaft that developed between 04:22 and 05:50 UTC (minutes 262-350 in Fig.
1 and 2). The shift in the location of the peak towards negative velocity near the cloud base (Fig. 5a) indicates the presence
of drizzle drops that dominate the radar signal. Gates near the cloud top on the other hand have peaks centered around the
zero velocity, indicating the presence of cloud drops. It is also noticeable in Fig. 5a the increase in the power of the signal as
drizzle drops become the dominant contribution to the radar reflectivity. To separate the drizzle from the cloud contribution
in the power spectra the assumption was made that the signal originating near the cloud top is *mostly* generated by cloud
droplets. This assumption holds true in weak and moderate drizzling conditions however fails in heavily precipitating clouds
when the Doppler spectra at the cloud top are also negatively skewed as the one at the cloud base. The spectra for layers near
the cloud top were vertically averaged and fitted to a Gaussian distribution. The standard deviation of the near-cloud-top
Gaussian distribution was taken as representative of the velocity spread of the cloud droplet distribution through the cloud.
Cloud-only spectra near the cloud top at 04:29, 04:35, and 04:56 UTC are shown in blue in Fig. 5 (b, c, d). Note that the
vertical velocity was converted into drop diameter using the relation between fall velocity and diameter from Frisch et al.,
(1995) and Gossard et al., (1990). To isolate the cloud component, the right shoulder of the curve is fitted to a Gaussian
distribution with standard deviation given by the cloud-only distribution (red curve). When this estimated cloud component
is subtracted from the cloud-averaged spectra, the resulting distribution (shown in yellow) is considered representative of the
drizzle-only signal. The areas under the final cloud and drizzle spectra (indicated by the red and yellow stripes respectively)
are proportional to the total mass of cloud and drizzle liquid water responsible for the radar signal. Although the analysis is
qualitative, it can be seen that the procedure captures the evolution of the drizzle from its initial stage to a stage where the
drizzle component becomes more prominent in the cloud.





## 3.2 Radar and radiometer

The areas under the red and yellow curves shown in Fig. 5 (b, c, d) are shown in Fig. 6 a, b for two entire shafts (04:22–05:50 UTC and 21:41–22:24 UTC). The radiometer-retrieved cloud and in-cloud drizzle LWP (black and red lines in Fig. 6 c, d) follow a similar time evolution. The missing points are times when the retrieval failed to converge. It should be noted that, as explained is Sect. 2, the drizzle water path below cloud base derived by the active module is used, together with an initial estimate of total LWP, to estimate the a priori partition between cloud and drizzle water path. During the retrieval

process the algorithm adjusts the PWV, total, and cloud LWP ($C_f$) to achieve convergence based on the microwave radiometer measurements. During this process both the cloud water and in-cloud drizzle water path are adjusted. Therefore, a correlation between the radar information and the radiometer retrieval is expected. Fig. 6 shows that the retrieval process conserves the information provided by the radar and, while adjusting the total liquid water path to be consistent with the scattering properties of the hydrometeors, it provides final estimates of in-cloud LWP that are consistent with the radar in-

cloud information and with the radar-provided retrievals below cloud base. In the two examples below, the radar and radiometer both show that the cloud LWP component is dominant through the shaft and the in-cloud drizzle increases to reach a maximum after about 10 minutes. The retrieved LWP in these two shafts shows that during the times of maximum drizzle development the in-cloud drizzle LWP reaches at the most 10-15% of the cloud LWP. The quantification of the drizzle LWP in relation to the total and cloud LWP is examined in the next section.

## 4 Analysis of results and potential applications

In this section cloud and drizzle LWP derived during 10 days each of open cellular and closed cellular stratocumulus cloud conditions observed at the ENA site are analyzed and discussed. The purpose of this section is to evaluate whether the results are consistent with the current state of knowledge of open and closed cell systems (Wood et al., 2008) and to provide ideas for possible applications of these results to the study of turbulence, drizzle production, drizzle formation, and cloud-aerosol

interaction.

Before proceeding with the details of the drizzle and cloud water path partition some general features of the retrieval applied to the 10 open cell cases are shown. The open cell cases are selected as they contain larger drizzle drops leading to greater scattering of the microwave signal, however similar conclusions can be drawn for the closed-cell data. Fig. 7a shows the reduction in the uncertainty of $C_f$ (ratio of cloud LWP to total LWP) after the retrieval converges. The retrieval has a larger

impact in cases where the drizzle diameter below cloud base is larger than 200 μm as shown in Fig. 7a. A $C_f$ value of unity corresponds to no drizzle drops within the cloud layer, and a value of zero corresponds to absence of any cloud sized drops in the cloud layer. The final retrieved $C_f$ varies between 0.5 and 1 (no drizzle) and is shown in Fig. 7b vs the a priori $C_f$ for clouds with LWP greater than 150 g/m$^2$. Collectively Figure 7a and 7b demonstrate the reduction in the uncertainty of $C_f$ due to the retrieval process.





As already mentioned in Sect. 2.1 the a priori total liquid water path ($LWP_t$), also used to start the convergence process, is derived with a neural network algorithm (Cadeddu et al., 2009) with no-scattering assumptions. The present retrieval generally reduces the $LWP_t$ with respect to the a priori and the reduction is more pronounced for cases affected by scattering to a larger extent (Fig. 7c). However, for a better understanding of the overall impact of the scattering effect on the total LWP, the present retrievals were compared with the results from MWRRETv2. The MWRRETv2 utilizes an optimal

estimation algorithm very similar to the one used in this work, but with an absorption-only radiative transfer model. Figure 7d shows that accounting for scattering effects reduces the total liquid water path by about 10-15% depending on the drizzle diameter. This result provides a quantification of the uncertainty that can be expected from neglecting scattering effects during precipitating conditions. For thicker clouds with $LWP_t > 500$ g/m$^2$, neglecting the scattering effects of drizzle drops when using the 90 GHz channel can potentially lead to an overestimation of LWP by ~100 g/m$^2$, far higher than the accuracy

needed for characterizing the aerosol-cloud interactions.

Tables 2 and 3 display a summary of the average cloud and drizzle characteristics in the shafts for each open cell and closed cell day analyzed. As expected, the total LWP is larger in the open cell cases compared to the closed cell, even accounting for the retrieval underestimation due lack of convergence during times with the highest precipitation. The co-variability of the total (in-cloud + below-cloud) DWP and the cloud LWP is explored in Fig. 8. Shaft averaged values of DWP and cloud

LWP are binned in bins centered at 50, 150, 250 and 350 g/m$^2$ with a width of 100 g/m$^2$. The total (in-cloud + below-cloud) drizzle water path in the shaft is a small fraction (generally less than 30%) of the cloud LWP and increases with the cloud water path. This behavior is consistent with the findings of Lebsock et al. (2011). The DWP increase is more pronounced in the open cell (shown in black) than in the closed cell (shown in red) systems, and for a similar amount of cloud LWP greater amount of drizzle is present in the open cellular shafts.

From Tables 1 and 2 it is evident that in closed cell cases 70-80% of the total drizzle is found in the cloud and less than 30% of the total drizzle in a shaft is precipitating. This contrasts with the open cell cases where on average 30-50% of the total drizzle is precipitating. This is further examined in Fig. 9 where the cumulative distribution of the ratio of precipitating-to-total drizzle water path in the shaft is shown segregated by the average drizzle diameter at the cloud base. The figure shows that the fraction of drizzle water path leaving the cloud is higher in shafts that, on average, have larger droplets. Virtually all

closed cell cases (blue line) have a drizzle diameter less than 200 microns (GC19) and for 90% of them the fraction of drizzle water path below the cloud is less than 0.2. In the same range of drizzle diameter open cell shafts (black line) show higher precipitation fraction with 90% of the shafts having below-cloud to total ratio of 0.4 or less. Finally, in 80% of the shafts with larger average drop sizes (red line) the ratio of below-cloud to total drizzle water path is 0.6 or less.

The partition of cloud and drizzle water path is also important when studying the relation between turbulence and

precipitation. As an example, Fig. 10 shows the total (a), below-cloud-base (b), and above-cloud-base (c) drizzle water binned by the radiative flux divergence at the cloud top and by total LWP for all 1-min averaged closed cell cases. The figure illustrates the relation between drizzle, LWP, and turbulence. Clouds with strong divergence (less than -100 W/m$^2$) have high probability of developing drizzle in the cloud when the LWP is above ~150 g/m$^2$. However, from Fig. 10





precipitation doesn't develop until the LWP is above ~200 g/m². The differences in the values of DWP below and above the

cloud base for a similar amount of radiative flux divergence at the cloud top and total LWP suggests drizzle might be present within the cloud before it is detected below the cloud base. In addition, the amount of drizzle water within the cloud is greater than the amount below the cloud base for almost all values of radiative cooling and LWP.

## 5 Summary and conclusions

In this work Mie scattering by larger drizzle drops in the microwave spectrum is exploited to partition cloud and drizzle

liquid water path using data from active and passive sensors. Brightness temperature observations from a microwave radiometer, profiles of lidar backscatter and profiles of the first three moments of the radar Doppler spectra serve as an input to the retrieval algorithm. These data together with a radiative transfer code that includes Mie scattering calculations are used to derive parameters of drizzle DSD below the cloud base, total column LWP, and cloud and drizzle water path above the cloud base in marine boundary layer clouds. Due to the lack of coincident observations of above-cloud DWP via aircraft

measurements, the retrieved cloud and drizzle water path above the cloud base during one day are qualitatively compared with the radar Doppler spectra between cloud base and cloud top. The analysis suggests that the optimal estimation algorithm utilizes the information provided by the radar and ceilometer on the drizzle below the cloud base to adjust the cloud liquid water path and in-cloud drizzle water path to achieve convergence. The converged solution is broadly consistent with the partition between cloud and in-cloud drizzle water path extracted from the radar Doppler spectra.

The retrieval algorithm is applied to 20 days of precipitating stratocumulus cloud conditions at the ARM ENA site. Quantitative analysis of the cloud and drizzle water path during 20 days of precipitating events at the ENA site shows differences between closed and open cell scenarios. In the closed cell systems, only a small fraction (~20%) of the available drizzle in the cloud falls below the cloud base as compared to the open cell (~40%). Precipitation is associated with strong radiative cooling at the cloud top (less than -100 W/m²) and higher liquid water path (higher than 200 g/m²). However,

drizzle in the cloud begins to exists at weak radiative cooling (divergence is less than -80 W/m²) and liquid water path higher than ~150 g/m². The amount of available drizzle that precipitates below the cloud base is higher (30-50 %) in open cell systems than in closed cell systems and is related to the average drizzle drop size. The average total drizzle water path in open cell shafts was fairly high, in all cases analyzed here it was higher than ~30 g/m² accounting for at least 20% of the total liquid water path retrieved by the radiometer. As the algorithm didn't converge during the highest precipitating intervals

of the open cell shafts it is reasonable to conclude that the estimates provided here are in certain cases an underestimation. Additionally, smaller drizzle drops in the cloud are undetected because their scattering effect is negligible in the microwave leading to a possible underestimation of the cloud DWP even in closed cell systems.

Our results primarily highlight the need to account for scattering by drizzle drops while retrieving the column amount of liquid water (LWP) from the brightness temperatures observed by high frequency microwave radiometers. Precipitation is

ubiquitous in marine stratocumulus clouds with much of it evaporating before reaching the surface (Zhou et al. 2015;



Remillard et al. 2012; Serpetzoglou et al., 2008). The LWP can be inaccurate by traditional (satellite and ground-based) algorithms that neglect the scattering due to drizzle drops for clouds with LWP greater than 500 $g/m^2$. This can lead to inaccurate quantification of adiabaticity (e.g. Kim et al., 2003; Kim et al., 2008), precipitation susceptibility (e.g. Sorooshian et al. 2009), and aerosol-cloud interactions (e.g. McComiskey et al. 2009). LWP is also one of the primary metrics for

evaluating single column model simulations and Large Eddy Simulation (LES) model in stratocumulus cloud conditions (e.g. Remillard et al. 2017; McGibbon and Bretherton, 2017). The DOE ARM program has had a strong impact on furthering our understanding of aerosol-cloud-precipitation interactions (Feingold and McComiskey, 2016) and on cloud modeling at various scales (Kruger et al. 2016; Randall et al. 2016). Although preliminary, our analyses have impact on the conclusions of some of the previous studies. Objective quantification of the overestimation of the LWP by the traditional algorithms is a

warranted and will be topic of our further study.

## 5 Author contribution

M.C. prepared the manuscript with contributions from all authors. V.G. preprocessed, cleaned and calibrated the radar and ceilometer data. M.C. performed the active and passive retrievals. M.M. contributed to the development and compilation of PAMTRA and provided support to the use of the PAMTRA radiative transfer model.

## 5 Acknowledgements

VG was supported by the U.S. Department of Energy's (DOE) Atmospheric System Research (ASR), an Office of Science, Office of Biological and Environmental Research (BER) program, under Contract DE-AC02-06CH11357 awarded to Argonne National Laboratory. MC is supported by the U.S. Department of Energy, Office of Science, Office of Biological and Environmental Research, Atmospheric Radiation Measurement Infrastructure, under contract # DE-AC02- 06CH11357.

The ground-based data used in this study were obtained from the Atmospheric Radiation Measurement (ARM) user facility, a U.S. Department of Energy (DOE) Office of Science, user facility managed by the Office of Biological and Environmental Research.

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






**Table 1: The passive module of the retrieval algorithm. Mean values and standard deviations of a priori; retrieved quantities for all the cases where the retrieval converged are shown in red.**

| Step | Variable | Initial Estimation | |
|---|---|---|---|
| First guess water vapor | PWV [kg/m$^2$] | Statistical retrieval (*) 1.63±0.35; 1.59±0.36 | |
| First guess total LWP | LWPt [g/m$^2$] | Statistical retrieval (*) 114.1±136.7; 92.9±103.5 | |
| | | **Below-cloud base** | **In-cloud** |
| Average drizzle effective radius | D0d [μm] | Active retrieval 159.3±103.5 | Constant=D0mix at cloud base 61.2±48.5 |
| Cloud effective radius | D0c [μm] | | Assumed = 20 |
| First guess drizzle LWC | DWC | Active retrieval | Constant = LWCmix at cloud base (**) |
| First guess drizzle LWP | DWP [g/m$^2$] | Integrated from DWC$_{bc}$ 6.4±12.7 | Integrated from DWC$_{ac}$ (**) 13.9±33.4; 10.4±24.9 |
| First guess cloud LWP | CWP [g/m$^2$] | | CWP=LWPt-DWP$_{ac}$ (**) 100.3±114.8; 82.6±88.9 |
| First guess cloud LWC | CWC | | Assumed adiabatic (**) |
| First guess cloud to total LWP ratio | C$_f$ | C$_f$=CWP/LWPt (*) 0.86±0.12; 0.92±0.15 | |

(*) Retrieved with passive module
(**) Adjusted during the retrieval to be consistent with integrated amounts






**Table 2: Cloud, drizzle, and total LWP, for open cell cases (units are g/m².).**

| Date | # shafts (min) | Total LWP | Below Cloud DWP (fraction of total DWP) | Above Cloud DWP** (fraction of total DWP) | DWP | CWP |
|---|---|---|---|---|---|---|
| 20151207 | 8 (199) | 303.51 | 24.98 (.38) | 41.72 (.62) | 66.70 | 236.81 |
| 20151230 | 4 (143) | 172.94 | 15.09 (.43) | 20.31 (.57) | 35.40 | 159.66 |
| 20160113 | 10 (286) | 214.49 | 15.66 (.40) | 23.07 (.60) | 38.72 | 176.66 |
| 20160329 | 9 (274) | 152.87 | 12.02 (.42) | 16.46 (.58) | 28.48 | 143.75 |
| 20160411 | 11 (285) | 135.51 | 15.39 (.58) | 11.05 (.42) | 26.44 | 122.54 |
| 20160508 | 8 (311) | 182.07 | 13.20 (.42) | 18.22 (.58) | 31.41 | 151.39 |
| 20160509 | 9 (237) | 128.20 | 10.74 (.39) | 16.89 (.61) | 27.63 | 117.87 |
| 20161022 | 12 (274) | 212.72 | 20.38 (.55) | 16.56 (.45) | 36.95 | 185.96 |
| 20161104 | 5 (158) | 174.66 | 10.37 (.31) | 22.69 (.69) | 33.05 | 141.61 |
| 20161121 | 13 (434) | 233.95 | 15.92 (.27) | 43.05 (.73) | 58.97 | 174.98 |
| All | 89 (2651) | 194.68±158.27 | 15.84±19.02 (.40) | 23.3±26.96 (.60) | 39.15±35.11 | 162.11±131.98 |



**Table 3: Cloud, drizzle, and total LWP, for closed cell cases (units are g/m$^2$).**

| Date | #shafts (min) | Total LWP | Below Cloud DWP (fraction of total DWP) | Above Cloud DWP (fraction of total DWP) | DWP | CWP |
|---|---|---|---|---|---|---|
| 20151019 | 3 (97) | 210.8 | 2.25 (.17) | 12.14 (.83) | 14.57 | 196.26 |
| 20160227 | 5 (417) | 138.06 | 4.47 (.20) | 18.76 (.80) | 23.40 | 114.66 |
| 20160303 | 3 (97) | 183.34 | 0.49 (.15) | 3.04 (.85) | 3.54 | 179.80 |
| 20160304 | 3 (212) | 215.57 | 1.20 (.14) | 8.27 (.87) | 9.49 | 206.08 |
| 20160409 | 10 (492) | 158.87 | 3.68 (.25) | 11.52 (.75) | 15.27 | 143.60 |
| 20160628 | 9 (550) | 123.49 | 3.46 (.20) | 15.42 (.80) | 19.25 | 104.25 |
| 20161015 | 5 (439) | 143.53 | 6.73 (.23) | 23.93 (.77) | 30.72 | 112.81 |
| 20161031 | 13 (575) | 158.20 | 3.53 (.24) | 11.02 (.76) | 14.57 | 143.63 |
| 20161116 | 8 (368) | 212.43 | 8.96 (.16) | 29.09 (.84) | 34.46 | 177.98 |
| 20161117 | 8 (436) | 129.96 | 9.92 (.29) | 23.87 (.71) | 33.53 | 96.42 |
| All | 65 (3603) | 159.95±56.20 | 4.97±5.32 (.22) | 16.31±14.38 (.78) | 20.91±18.21 | 139.05±49.88 |





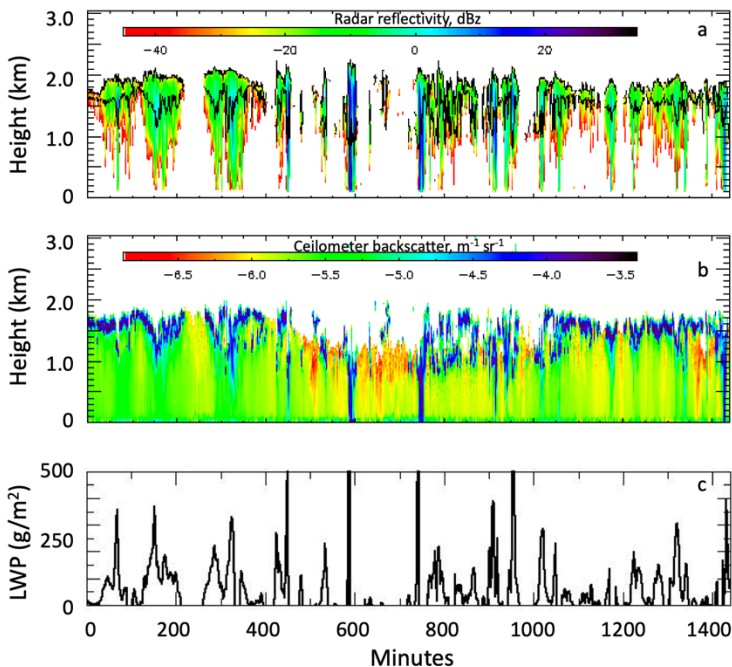

**Figure 1: (a)** Time-height profiles of KAZR reported reflectivity (shades), ceilometer reported cloud base and top height (black), **(b)** time-height profiles of ceilometer reported backscatter (shades), and **(c)** time-series of microwave radiometer reported LWP from MWRRETv2. The data were collected on 21 November 2016. Data in a and b are 1-minute averaged, data in c are smoothed with a 5-min running average.


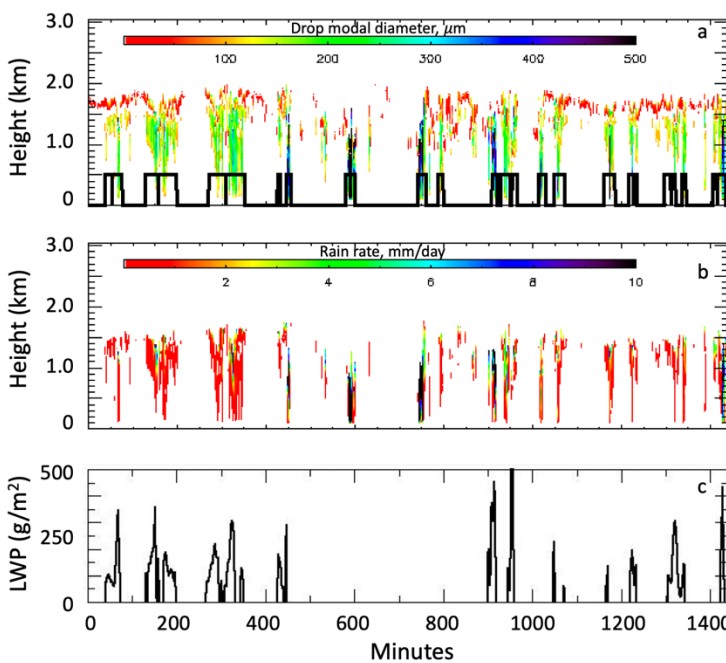

**Figure 2: (a)** Time-height profiles of retrieved drizzle drop diameter (shades) and identified drizzle shafts (black line) **(b)** time-height profiles of rain rate and **(c)** time-series of retrieved LWP during precipitating shafts using PAMTRA. The data were collected on 21 November 2016. Data in a and b are 1-minute averaged, data in c are smoothed with a 5-min running average.










**Figure 3: Flow chart of the active and passive modules.**





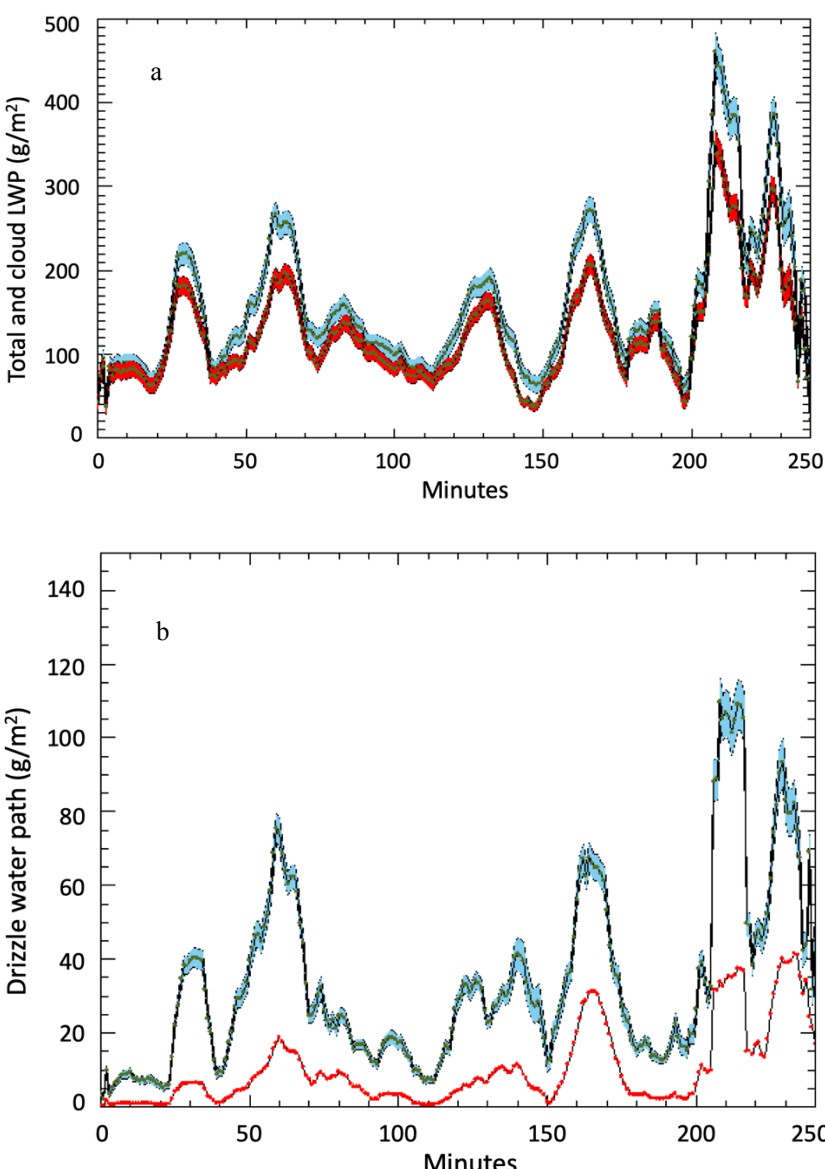

**Figure 4: (a) Total (blue) and cloud (red) LWP. (b) Total drizzle water path (blue) and below-cloud drizzle water path (red) between 00 and 04 UTC on 21 November 2016. The data are smoothed with a 10-min boxcar average for better readability.**
**Shaded regions represent the 1-sigma uncertainty provided by the optimal estimation algorithm.**





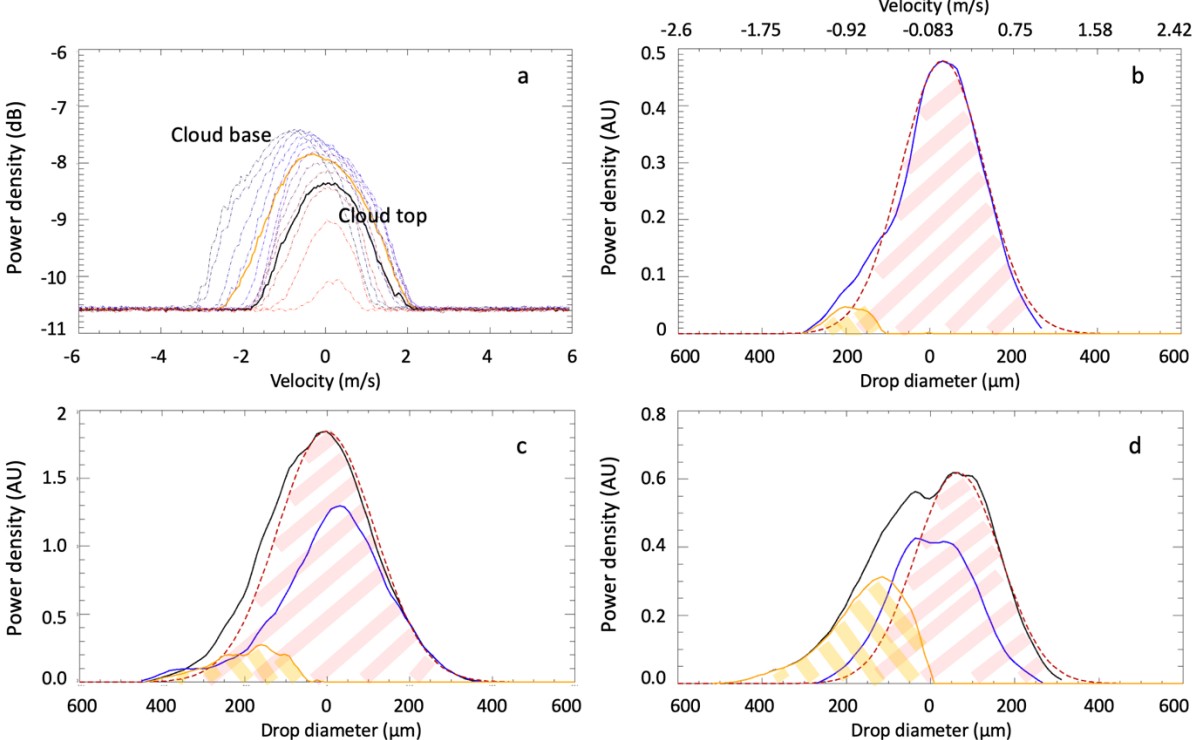


**Figure 5: (a) Radar Doppler spectra between cloud base and cloud top at 04:30 UTC on November 21, 2016. (b–d) Doppler spectra averaged between cloud base and cloud top (black), averaged over cloud-only layers (blue), Gaussian fitted curve (red), drizzle component (yellow) at 04:29 (b), 04:35 (c), and 04:56 (d) UTC. All Doppler spectra are minute averaged. On the top axis the velocity corresponding to the calculated diameter is shown. Negative velocities refer to down motion.**





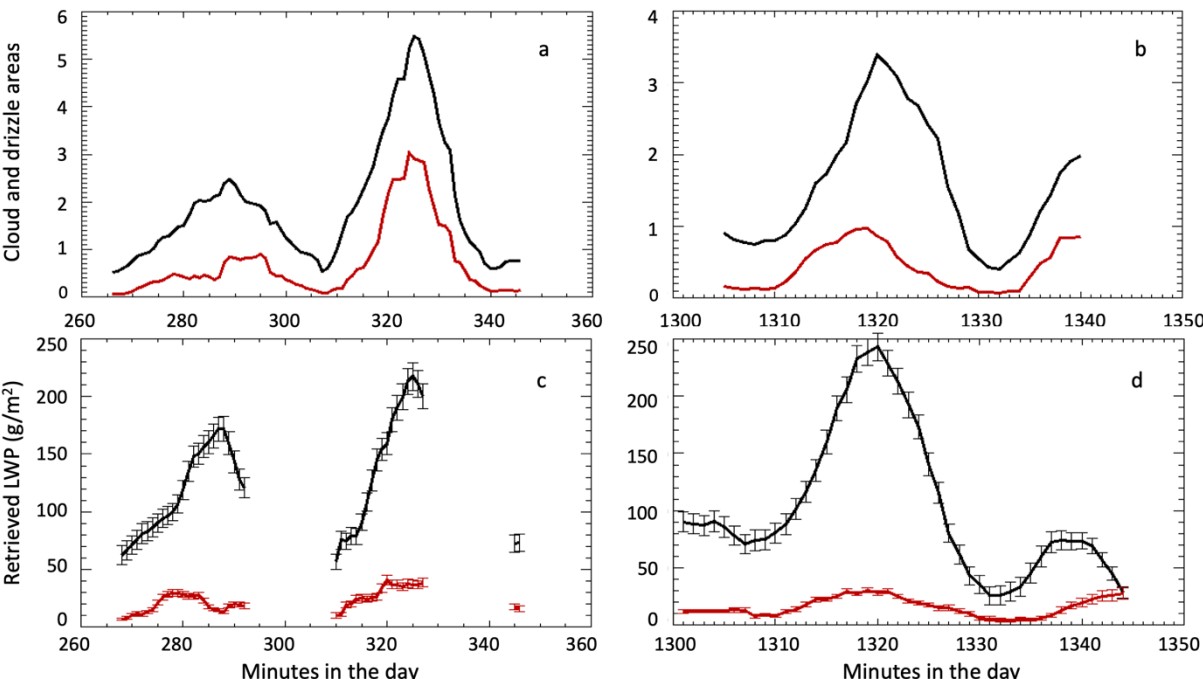


**Figure 6: Cloud (black) and drizzle (red) areas derived from Doppler spectra during two precipitating shafts at 4:22–5:50 UTC (a) and 21:41–22:24 UTC (b) on November 21, 2016. In the bottom panels corresponding cloud LWP (black), and in-cloud drizzle water path (red) estimated by the passive module are shown for the same shafts (c, d).**







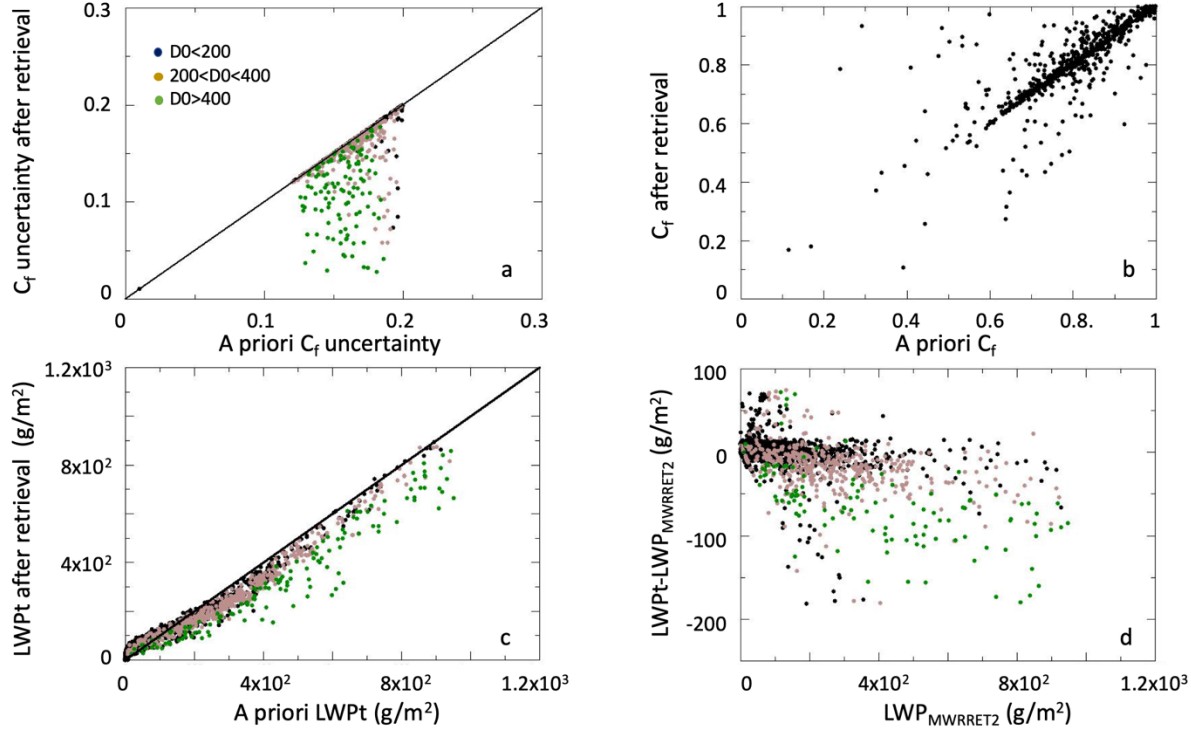

**Figure 7: (a) A priori $C_f$ uncertainty (X-axis) and $C_f$ uncertainty after the retrieval (Y-axis); (b) A priori $C_f$ (X-axis) and $C_f$ estimated after the retrieval (Y-axis) for samples with LWP greater than 150 g/m²; (c) A priori total LWP (X-axis) and LWP**
**estimated after the retrieval (Y-axis); (d) Difference between retrieved total LWP and the LWP retrieved with MWRRET2 that uses an optimal estimation approach with absorption-only radiative transfer model. In panels c and d, only samples with LWP < 1000 g/m² are shown.**








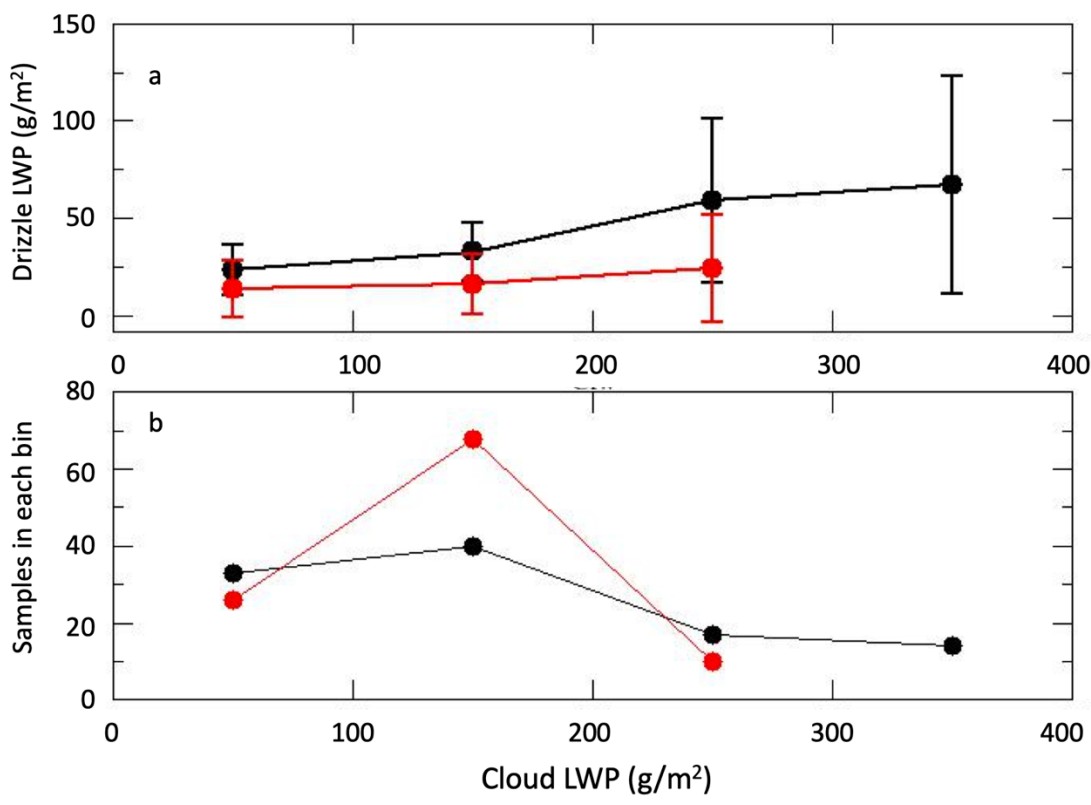

Figure 8: (a) Mean and standard deviation of cloud and drizzle LWP for open cell (black) and closed cell (red) shafts. (b) Number of samples in each bin for open cell (black) and closed cell (red) shafts.





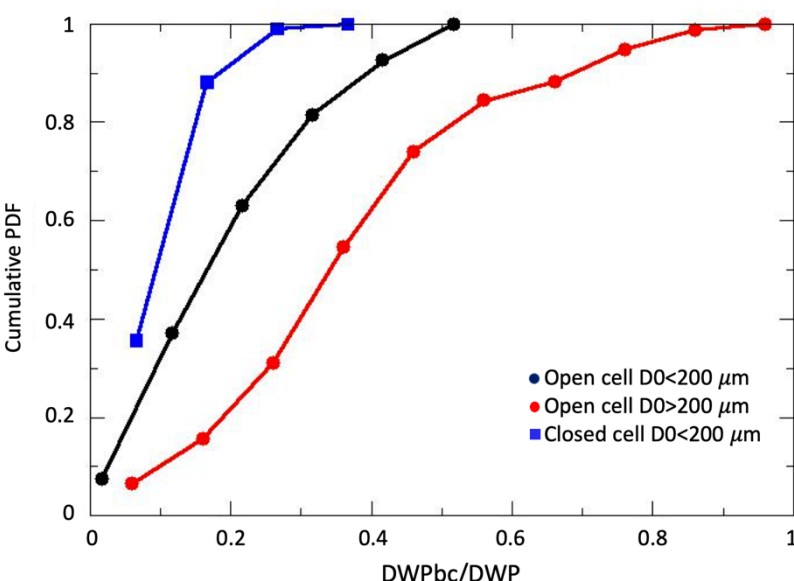

**Figure 9: (a) Cumulative distribution of below-cloud to total drizzle water path for open and closed cell cases segregated by drizzle modal diameter (D0) at the cloud base.**








**Figure 10: (a) Total, (b) below cloud base, and (c) in-cloud drizzle water path binned by radiative divergence and total liquid water path. The solid line represents the total LWP binned by flux divergence.**