# Peer review of "Ground-based Observations of Cloud and Drizzle Liquid Water Path in Stratocumulus Clouds"

_Atmospheric Measurement Techniques, 2019_

## Referee Comment (RC1) · Anonymous Referee #2 · 18 Dec 2019

The paper by Cadeddu presents a new technique to retrieve column integrated values of drizzle water below and above cloud base as well as cloud water above the cloud base.

The technique is well presented, but is only applied only to a small data set. However, the paper fails to provide necessary information to evaluate if the technique can be applied, for example, only to geometrical thin clouds or only to warm clouds. I would be good to know the range of, e.g. cloud optical and geometrical thickness or cloud top temperature of the clouds that can be considered as potential targets for the technique. The authors should also state if the technique only works for single cloud layers or how

the observed LWP would be distributed over multi-layered clouds. Can the method could also be applied to Arctic clouds?

I would be very helpful if the authors would provide a brief review on cloud-droplet size distributions and drizzle size distributions. What are typical values in the literature for warm stratocumulus clouds? The calculated cloud droplet diameters shown in Figure 5 seem quite large and the drizzle diameters rather small.

Minor comments:

Line 87: calculations are based . . . for non-spherical and oriented particles. How are spherical droplets (cloud/drizzle) handled in the model?

Line 89: what do you understand under ice crystal habit?

Line 212: What drizzle size was observed? Please add a figure of the observed DSD in cloud and below cloud for the different cases and add in Table 2 and 3 the mean cloud and drizzle (in and below cloud) diameter, CTT, and optical and geometrical thickness.

Figure 1: add the observed precipitation at ground

Figure 2/line 109: Drizzle modal diameter is not shown in the Figure 2. Please change. Also, change the colour scale, maybe use a log scale. Now it is only shown to 500 $\mu$m. It should be extend to the 800 $\mu$m (largest diameter stated in the text).

Figure 5b, black line is missing.

Other comments: Figure 5, yellow is not a good choice of colour. The contrast is very poor.

Figure 7, The colour in the legend and the plotted data seem not to be the same.

[Figure]

---

## Referee Comment (RC2) · Anonymous Referee #1 · 19 Dec 2019

General comments

This manuscript presents a new technique for obtaining cloud and drizzle liquid water path by combining multi-channel microwave radiometer, Doppler cloud radar and ceilometer measurements. The new technique is applied to observations of precipitating stratocumulus clouds and evaluated qualitatively by comparison with Doppler cloud radar spectra.

The technique shows great promise and will aid the community investigating the properties of stratocumulus by providing a new piece of information, although the full potential is not explored deeply in this initial study. This manuscript is almost ready for

publication, with a few technical aspects to correct.

Technical comments

Line 18: Replace 'exists' with 'exist'.

Line 22: Suggest opening with 'Marine stratocumulus clouds have a significant impact on the Earth's radiation balance as they reflect a greater amount of solar radiation back to space compared to the ocean surface, and emit a similar amount of longwave radiation as the surface.'

Line 26: Replace 'Feingold and A. McComiskey 2016' with 'Feingold and McComiskey 2016'.

Line 31: Move comma from after 'properties' to after 'instrumentation' .

Line 32: Do you mean moments here, or would it be more realistic to state 'the shape of the drop size distribution'? Otherwise you should explain what you mean by moments in this context.

Lines 35-36: This statement needs some qualification. Review papers discussing LWP estimation from multi-channel microwave radiometers usually state that care must be taken in the presence of precipitation, and that LWP estimates are not reliable in strong precipitation.

Line 58: Replace 'since summer of 2015' with 'since the summer of 2015'.

Line 61: Suggest stating 'reflectivity-weighted Doppler spectrum'

Line 61: Replace 'Collocated to' with 'Collocated with'.

Line 63, 75, 92 and elsewhere: Replace 'backscatter' with 'attenuated backscatter'.

Line 71: The correct reference for 'auto-calibration of cloud lidar' is O'Connor et al. (2004) not (2005).

Line 103: Lidar ratio for cloud droplets at 905 nm is about 19 sr, and is even lower for

larger drizzle droplets.

Line 104: The ceilometer attenuated backscatter peaks at cloud base due to the large return from the small but much more numerous cloud droplets, relative to drizzle droplets.

Line 106: Do you mean here, 'the average modal diameter of the full drop size distribution including drizzle drops and cloud droplets'? How reasonable is this assumption considering that these are two distinct hydrometeor populations, normally giving rise to a skewed distribution if they overlap?

Line 161: Small drizzle drops may not display a negative Doppler velocity if they are falling into a strong updraft. It is true to state that drizzle drops have a significant terminal fall velocity, but the observed Doppler velocity is the sum of the fall velocity and the air motion.

Line 167: Suggest using the term 'drizzle shafts' here and elsewhere in the manuscript.

Line 176: Suggest rephrasing to '.. are as negatively skewed as the Doppler spectra at cloud base'.

Lines 177-178: The terminal fall velocity of cloud droplets is very small, and their observed Doppler velocity distribution is a result of turbulence.

Line 185: Not quite true. For Rayleigh scattering, reflectivity is proportional to mass-squared, but the larger drizzle drops are in the Mie scattering regime.

Line 264: Do you mean in-cloud DWP here?

Figure 1: 'together with cloud boundaries from KAZR (cloud top) and ceilometer (cloud base)' 'ceilometer attenuated backscatter coefficient'

Figure 4: In (a), does cloud LWP include in-cloud drizzle (DWP) or cloud droplet LWP only?

Figure 5: 'Downward motion'. It is not clear how the x-axis is derived.

Figure 10: The solid line represents the mean of the total LWP measured in each flux divergence bin? How about the bars? The figure caption should be clear.

---

## Referee Comment (RC3) · Anonymous Referee #3 · 20 Dec 2019

Review of 'Ground-based Observations of Cloud and Drizzle Liquid Water Path in Stratocumulus Clouds' by Cadeddu et al.

This manuscript presents a new methodology for retrieving simultaneously the cloud liquid water path as well as the drizzle water path below and in the cloud layer. This is achieved by combining active and passive ground-based instrumentation. This work is largely based on previous research by the authors and the foundations for this retrieval algorithm have therefore already been evaluated. The novelty consists in the way active and passive measurements are merged to infer more precise in-cloud information. Further analyses are provided, in particular concerning the importance of scattering effects for such retrievals. Finally, 10 days of retrievals for open and closed-cell marine stratocumuli are statistically analysed.

The manuscript is well written, and the retrieval technique (although relatively complex) seems promising. I do not see any particular issue with the retrieval technique, although a main issue is of course the difficulty to evaluate its results. The authors here offer quantitative evaluations through comparisons with radar Doppler spectra or with expectations based on previous literature results. It might be too limited to make strong conclusions on the cloud and drizzle properties in open/closed cell stratocumuli, but I acknowledge it is difficult to go further in the absence of in situ measurements and without more statistics than the 10 days analyzed here. Overall, I think this is convincing and interesting work, and would suggest for publication after minor revisions, following the comments below.
* * *
General comments:

1. One of the main findings of this paper concerns the importance of considering scattering effects in microwave retrievals. However, I have a few concerns with the ways these results are obtained.

A first way to evaluate the scattering effect is through comparisons between the retrievals and their associated a priori values from a neural network algorithm that doesn't consider scattering. I am not very convinced with the impact on $C_f$, which seems to stay close to its a priori value, but a reduction of LWPt is indeed clearly observed. Is the optimal estimation framework used for this study based on a Levenberg-Marquardt scheme, i.e. is a departure from the a priori value actually showing a reduction of the cost function (rather than being possible iteration noise in a Gauss-Newton approach)? Please comment on this, and for future study I'd suggest using more quantitative metrics like the cost function, information content or degrees of freedom to reach such conclusions.

Another way the importance of scattering is quantified is by comparing the retrievals of the new technique to those of MWRRET2, a similar retrieval algorithm. Considering the importance of these results, more details of the similarities / differences between the retrieval algorithms should be given in section 2.1. But why not simply turn scattering off (forcing the single-scattering albedo to 0) in your current retrieval algorithm, instead of using a different retrieval algorithm? That would avoid being impacted by retrieval technique differences and be much more convincing.

2. The impact of shafts on retrievals is discussed, first in the algorithm description and then in the result discussions. But it is still not clear to me, especially in the discussions surrounding

Tables 2 and 3, what part of the conclusions concern impacts from retrieval limitations or from actual microphysics differences during shafts. Please clarify the exact (expected) impact of shafts on retrievals, so that the readers can more clearly understand your results.

Specific comments:

3. The title could be slightly more specific to reflect better the main findings of this study (e.g. concerning the in-cloud drizzle retrievals, or the importance of scattering).

4. p3 l87-88: Do I understand correctly that non-spherical and oriented particles models are used to describe cloud droplets and drizzle? Is there any particular reason?

5. p4 l98: I think it would be worth expliciting the DSD and its 3 parameters, as different DSD shapes are used in the literature.

6. p4 l105-107: Dealing with a mixtures of cloud droplets and drizzle within the same pixel clearly is a challenge. Is there a reference to justify the choice of a lognormal shape with fixed width?

7. p7 l194: typo: "in"

---

## Author Comment (AC1) · 30 Jan 2020

We thank the reviewer for very detailed comments on the manuscript. Responding to them has substantially improved the manuscript. The reviewer's comments are italicized and responses are in red.

*The paper by Cadeddu presents a new technique to retrieve column integrated values of drizzle water below and above cloud base as well as cloud water above the cloud base.*
*The technique is well presented, but is only applied only to a small data set. However, the paper fails to provide necessary information to evaluate if the technique can be applied, for example, only to geometrical thin clouds or only to warm clouds. I would be good to know the range of, e.g. cloud optical and geometrical thickness or cloud top temperature of the clouds that can be considered as potential targets for the technique.*

To address this concern, we added more discussion on this in section 5 at lines 417-428. Rather than specific atmospheric conditions under which the technique can be used, below we report specific criteria under which the technique can be applied,
1) The radar and ceilometer are not attenuated by precipitation and are able to adequately detect the cloud base and cloud top.
2) The radiometer measurements are not affected by precipitation on the lens.
3) The drizzle droplet diameter is large enough to be detected by the 90 GHz channel (in other words the technique will not work in very light drizzle).
4) The cloud can be considered close to be adiabatic so that the cloud and in-cloud drizzle water content can be modeled with sufficient confidence.

Given these criteria the applicability of the technique can be different for ground-based and airborne instrumentation, and for a combination of the two. For example, if we had a radiometer looking down instead of looking up the criterion #2 would be satisfied for a broader range of precipitating clouds than what was presented in this work, as long as the other criteria are met. The attenuation at Ka-band wavelength is significant during heavy precipitation, making it not possible to retrieve below-cloud cloud drizzle properties. The adiabaticity of marine stratocumulus clouds changes on shorter (less than minute) timescales, with sub-adiabatic downdrafts and super-adiabatic updrafts (Stevens et al. 1998, Wood, 2012). However, the clouds are nominally adiabatic on minute or longer timescales, suitable for application of this technique. We have also added in Tables 3-5 the estimated optical depths for the clouds in this work (assuming a cloud drop effective radius of 10 μm) and the geometrical thickness from the radar-estimated cloud top and the ceilometer-estimated cloud base. We think that the value reported are optimal for the application of this technique.

Stevens, B., W.R. Cotton, G. Feingold, and C. Moeng, 1998: Large-Eddy Simulations of Strongly Precipitating, Shallow, Stratocumulus-Topped Boundary Layers. J. Atmos. Sci., 55, 3616–3638, https://doi.org/10.1175/1520-0469(1998)055<3616:LESOSP>2.0.CO;2

*The authors should also state if the technique only works for single cloud layers or how the observed LWP would be distributed over multi-layered clouds.*

We used the technique for single layer clouds. In the open cell dataset examined for this work there were several occurrences of heavy precipitating stratocumulus clouds with non-precipitating shallow cumulus clouds in the layers below. These cases were usually heavy precipitating and therefore the passive retrieval was not applied. Theoretically, the technique could be applied to multi-layer clouds, however, as the reviewer mentioned here, a realistic representation of the cloud boundaries and LWP may be needed. This was added in section 5 lines 423-425.

*Can the method could also be applied to Arctic clouds?*

The falling ice/snow below a mixed phase Arctic clouds can potentially scatter the microwave radiation at 90 GHz emitted by the liquid water within the cloud. However, that will depend significantly on the shape and size of the ice crystals. This is outside the scope of this work and hence at this stage we can't recommend this methodology for Arctic clouds.

*It would be very helpful if the authors would provide a brief review on cloud-droplet size distributions and drizzle size distributions. What are typical values in the literature for warm stratocumulus clouds? The calculated cloud droplet diameters shown in Figure 5 seem quite large and the drizzle diameters rather small.*

Thank you for raising this issue. A comprehensive survey of cloud drop size distributions have been carried out by Miles et al. (2000) with estimates from multiple field campaigns reported in various articles e.g. DYCOMS-II Stevens et al. (2003 BAMS), VOCALS Zheng et al. (2011 ACP), Bretherton et al. (2010 ACP), EPEACE (Russell et al. 2013) and CSET (Albrecht et al. 2019). A comprehensive review of stratocumulus clouds is also provided in Wood et. al. (2011) and Wood (2012).
Due to the large variability of in-cloud and precipitation microphysical properties (diameter and number) both vertically and horizontally due to turbulence and aerosol-cloud interactions, many of these estimates are for bulk properties such as rain rates, LWC and LWP. Tables 3 and 5 now added to this work provide information of typical properties for these clouds.

Miles, N.L., Verlinde, J. Clothiaux, E. E.: Cloud Droplet Size Distributions in Low-Level Stratiform Clouds, *J. Atmos, Sci.*, 57, 295--311, 2000.

R. Wood, C. S. Bretherton, D. Leon, A. D. Clarke, P. Zuidema, G. Allen, and H. Coe: An aircraft case study of the spatial transition from closed to open mesoscale cellular convection over the Southeast Pacific, *Atmos. Chem. Phys.*, 11, 2341–2370, doi:10.5194/acp-11-2341-2011, 2011

Wood, R.: Stratocumulus clouds, *Mon. Weather Rev.*, 140, 2373--2423, 2012.

**Minor comments:**

*Line 87: calculations are based . . . for non-spherical and oriented particles. How are spherical droplets (cloud/drizzle) handled in the model?*

The single scattering properties of spherical droplets such as cloud, drizzle, or rain are calculated with the Mie theory. For the radiative transfer solver RT4 it doesn't matter if the particles are spherical or non-spherical. It simply gets the 4 by 4 scattering and extinction matrix and the emission vector as input. These have to be calculated or provided by appropriate methods.

*Line 89: what do you understand under ice crystal habit?*

By ice crystal habits we mean the shape, density, size, mass-size relation, and so on. All that which differentiates frozen particles in terms of radiative properties.

*Line 212: What drizzle size was observed? Please add a figure of the observed DSD in cloud and below cloud for the different cases and add in Table 2 and 3 the mean cloud and drizzle (in and below cloud) diameter, CTT, and optical and geometrical thickness.*

We added in Fig. 8, left panels (new) the distribution of the retrieved drizzle diameter below cloud base and what was retrieved immediately above cloud base with the radar only. Because the number of columns was too large to keep in one table, we added tables 3 and 5 with the shaft-averaged drizzle diameter found below and above cloud base, cloud top temperature, optical and geometrical thickness. Throughout the paper the cloud droplet diameter is assumed constant with a value of 10 micron. This value is also used in the calculations of the optical depth.

*Figure 1: add the observed precipitation at ground*

We added panel 1d with the precipitation at the ground observed by the video-disdrometer. Because of the large range of precipitation values the vertical axis is shown in log scale.

*Figure 2/line 109: Drizzle modal diameter is not shown in the Figure 2. Please change. Also, change the colour scale, maybe use a log scale. Now it is only shown to 500 μm. It should extend to the 800 μm (largest diameter stated in the text).*

Accepted. Thank you for this suggestion.

*Figure 5b, black line is missing.*

For this case which was at the very onset of the drizzle event the black line was entirely under the blue line. We state this in the caption now.

***Other comments:***
*Figure 5, yellow is not a good choice of colour. The contrast is very poor.*

Changed to green. Thanks.

*Figure 7, The colour in the legend and the plotted data seem not to be the same.*

Figure 7 was changed, and the colors were changed to be the same. Thanks.

---

## Author Comment (AC2) · 30 Jan 2020

***General comments***

*This manuscript presents a new technique for obtaining cloud and drizzle liquid water path by combining multi-channel microwave radiometer, Doppler cloud radar and ceilometer measurements. The new technique is applied to observations of precipitating stratocumulus clouds and evaluated qualitatively by comparison with Doppler cloud radar spectra.*

*The technique shows great promise and will aid the community investigating the properties of stratocumulus by providing a new piece of information, although the full potential is not explored deeply in this initial study. This manuscript is almost ready for publication, with a few technical aspects to correct.*

Thank you for the review and the kind words. We hope to perform a more extensive study in the near future on the impact of these results on aerosol-precipitation interactions. Our responses to your comments are below in red.

***Technical comments***

*Line 18: Replace 'exists' with 'exist'.* Accepted.

*Line 22: Suggest opening with 'Marine stratocumulus clouds have a significant impact on the Earth's radiation balance as they reflect a greater amount of solar radiation back to space compared to the ocean surface, and emit a similar amount of longwave radiation as the surface.'* Accepted.

*Line 26: Replace 'Feingold and A. McComiskey 2016' with 'Feingold and McComiskey 2016'.* Thanks for catching that. It has been replaced.

*Line 31: Move comma from after 'properties' to after 'instrumentation'.* Accepted.

*Line 32: Do you mean moments here, or would it be more realistic to state 'the shape of the drop size distribution'? Otherwise you should explain what you mean by moments in this context.*

We have replaced the sentence to read. "From the point of view of ground-based instrumentation, the study of microphysical and macro-physical cloud properties involves combining data from multiple instruments to retrieve parameters of the hydrometeor drop size distribution (DSD). For example, the radar reflectivity is proportional to the sixth moment of the DSD and was used to retrieve liquid water content that is the third moment of DSD by Frisch et al. (2002)."
Thanks.

*Lines 35-36: This statement needs some qualification. Review papers discussing LWP estimation from multi-channel microwave radiometers usually state that care must be taken in the presence of precipitation, and that LWP estimates are not reliable in strong precipitation.*

We agree with the reviewer and clarified this statement in lines 53-57. Added 2 references on the topic (Wall et al., 2017 and Bosisio et al., 2013).

*Line 58: Replace 'since summer of 2015' with 'since the summer of 2015'.* Done-Thank you

*Line 61: Suggest stating 'reflectivity-weighted Doppler spectrum'.* Done

*Line 61: Replace 'Collocated to' with 'Collocated with'.* Done

*Line 63, 75, 92* and elsewhere: *Replace 'backscatter' with 'attenuated backscatter'.* Done for all instances. Thanks for pointing this important difference.

*Line 71: The correct reference for 'auto-calibration of cloud lidar' is O'Connor et al. (2004) not (2005).* Done

*Line 103: Lidar ratio for cloud droplets at 905 nm is about 19 sr, and is even lower for larger drizzle droplets.* Changed

*Line 104: The ceilometer attenuated backscatter peaks at cloud base due to the large return from the small but much more numerous cloud droplets, relative to drizzle droplets.* Changed

*Line 106: Do you mean here, 'the average modal diameter of the full drop size distribution including drizzle drops and cloud droplets'? How reasonable is this assumption considering that these are two distinct hydrometeor populations, normally giving rise to a skewed distribution if they overlap?*

Yes, and we agree with the reviewer that this is not the optimal solution. This assumption was very much debated among the authors and we resorted to this option because there is really no sensible way of separating the two distributions. This assumption was only used in the passive retrieval as a way forward to constrain the drizzle size in the cloud. It may require a separate study to understand how optimal this assumption is. In a recent study Glienke et al. (2017) pointed out that the cloud and drizzle distributions are almost in a continuum in marine stratocumuli. However, as they are measured by separate in situ probes, and modelled through different processes, the cloud and drizzle DSD are often assumed to be separate.

Glienke, S., A. Kostinski, J. Fugal, R. A. Shaw, S. Borrmann, and J. Stith (2017), Cloud droplets to drizzle: Contribution of transition drops to microphysical and optical properties of marine stratocumulus clouds, *Geophys. Res. Lett.*, 44, 8002–8010, doi:10.1002/ 2017GL074430.

*Line 161: Small drizzle drops may not display a negative Doppler velocity if they are falling into a strong updraft. It is true to state that drizzle drops have a significant terminal fall velocity, but the observed Doppler velocity is the sum of the fall velocity and the air motion.*

Thank you for raising this point and we agree with the reviewer that drizzle drops falling in updraft will not fall and rather go upwards, and the radar reported mean Doppler velocity is the sum of the droplet fall velocity and the air motion.

However, we resorted to converting the Doppler velocity to diameter as i) the focus of this study is on (relatively) larger drizzle drops with diameters greater than 100 micrometers that scatter radiation from the cloud and have fall velocity of 0.3 m/s spanning six Nyquist velocity bins  ii) the Doppler spectra are averaged on minute timescales in an attempt to minimize the contribution

from turbulence, and iii) for the cases analyzed here we didn't encounter a Doppler spectra entirely on the positive velocity.

The sentence has been rephrased as follows: "The methodology is based on the fact that the Doppler spectra of a non-precipitating cloud is centered on zero mean velocity due to their movement with turbulence, while that containing falling drizzle drops is negatively skewed due to their fall velocity. Hence, the presence of drizzle drops in a cloud introduces a negative skewness in the cloud Doppler spectra."

*Line 167: Suggest using the term 'drizzle shafts' here and elsewhere in the manuscript.* Done.

*Line 176: Suggest rephrasing to '.. are as negatively skewed as the Doppler spectra at cloud base'.* Accepted.

*Lines 177-178: The terminal fall velocity of cloud droplets is very small, and their observed Doppler velocity distribution is a result of turbulence.* Added.

*Line 185: Not quite true. For Rayleigh scattering, reflectivity is proportional to mass- squared, but the larger drizzle drops are in the Mie scattering regime.*

The reviewer is correct that for large drizzle drops that are under Mie scattering regime, the radar reflectivity is not proportional to mass-squared. Our forward model calculations show the Mie-to-Rayleigh backscatter ratio to be 1 for diameters below 400 micrometers, increasing to 1.2 for diameters of 1000 micrometers at Ka-band wavelength (Ghate and Cadeddu, 2019 JGR).

For the drizzle drops analyzed here, we estimate a maximum error of 20% due to this assumption. Further, even under the Mie scattering regime the area under the curve of the Doppler spectra will be still proportional to the mass of the condensate, albeit with a different proportionality than square. We have rephrased the sentence as follows:

"The areas under the final cloud and drizzle spectra (indicated by the red and yellow stripes respectively) are proportional to the total mass of cloud and drizzle liquid water responsible for the radar signal under the Rayleigh scattering regime with some modifications during Mie scattering regime."

*Line 264: Do you mean in-cloud DWP here?* Yes, it was intended above cloud base, we changed it with "in-cloud".

*Figure 1: 'together with cloud boundaries from KAZR (cloud top) and ceilometer (cloud base)' 'ceilometer attenuated backscatter coefficient'.* Changed

*Figure 4: In (a), does cloud LWP include in-cloud drizzle (DWP) or cloud droplet LWP only?* It only includes cloud droplets.

*Figure 5: 'Downward motion'. It is not clear how the x-axis is derived.*

We used the relationship between size and velocity in Gossard et al., 1990: r=av+b with a=1.4E-4 and b=1E-5. This is now stated in the caption.

*Figure 10: The solid line represents the mean of the total LWP measured in each flux divergence bin? How about the bars? The figure caption should be clear.*

Thank you, that was forgotten. The caption was rephrased as follows: The black circles connected by a solid line represent the total LWP binned by flux divergence and the vertical bars represent the standard deviation of the data in each bin.

---

## Author Comment (AC3) · 30 Jan 2020

We thank the reviewer for a detailed review. Please find below our responses to your comments. Your comments are in black and our responses are in red.

1. *One of the main findings of this paper concerns the importance of considering scattering effects in microwave retrievals. However, I have a few concerns with the ways these results are obtained.*
*A first way to evaluate the scattering effect is through comparisons between the retrievals and their associated a priori values from a neural network algorithm that doesn't consider scattering. I am not very convinced with the impact on $C_f$, which seems to stay close to its a priori value, but a reduction of $LWP_t$ is indeed clearly observed. Is the optimal estimation framework used for this study based on a Levenberg-Marquardt scheme, i.e. is a departure from the a priori value actually showing a reduction of the cost function (rather than being possible iteration noise in a Gauss-Newton approach)? Please comment on this, and for future study I'd suggest using more quantitative metrics like the cost function, information content or degrees of freedom to reach such conclusions.*

Yes, the convergence is monitored through a reduction of the cost function and through a convergence criterion as explained in the 2017 paper (C2017) eq 4. Because the problem is fairly well defined the convergence is very quick. The aspects mentioned by the reviewer are very relevant and they are at the very heart of the problem. The main reason why they were not addressed in more details in this work is because they were analyzed in detail in C2017 and here we wanted to focus more on the application of the retrieval rather than the retrieval itself, and also not to repeat previous analysis. Nonetheless, given the importance of the topic we have expanded section 4 and included more references to the results from the 2017 paper.

As it can be seen in Fig. 6 and 8 and Table I of C2017 the $C_f$ showed a small improvement with respect to the a priori. The degrees of freedom were also analyzed in Fig. 7 where it was found that the DOF of the system for $C_f$ varied depending on the physical constraints on the system. In this work a few changes were made, in particular only 3 quantities are retrieved and the drizzle DSD is provided. The a priori information for $C_f$ is also better constrained because is derived with the help of the active retrieval. Although it is true that the change in $C_f$ is not large, it is also possible that the a priori information provided is in within the limits of what can be achieved with this technique. The a-posteriori uncertainty of this parameter shown in Fig. 7a (this work) does show a reduction.

We have now included in Fig. 7 c and d more details on the retrieval. Fig. 7c shows the third element of the averaging kernel matrix A(3,3) defined in Eq. 5 of C2017 in relation to the average drizzle diameter. Fig. 7d shows one example of convergence. In this case (as in the majority of the cases that were able to converge) convergence is achieved at the 3rd iteration. In Fig. 7d for example the $C_f$ parameter is quickly adjusted from 0.73 to 0.59. On the right axis of Fig. 7d we now show the cost function is shown. A(3,3) represents the varying contribution of the measurements to Cf that depends partially on the amount of scattering that the model attributes to the scene. The discrete values are due to the truncation of the DFS values to the first decimal digit.

*Another way the importance of scattering is quantified is by comparing the retrievals of the new technique to those of MWRRET2, a similar retrieval algorithm. Considering the importance of these results, more details of the similarities / differences between the retrieval algorithms should be given in section 2.1. But why not simply turn scattering off (forcing the single-scattering albedo to 0) in your current retrieval algorithm, instead of using a different retrieval algorithm? That would avoid being impacted by retrieval technique differences and be much more convincing.*

We thank the reviewer for this comment. This is actually something that was debated during the writing of the manuscript. The rationale for showing the comparison with MWRRET was that, because users are going to utilize MWRRET, they may be interested in knowing how much scattering is affecting that retrieval. However, we also see the point of the reviewer here and agree that using the same radiative transfer code

and methodology has its merits as it eliminates possible differences and biases due to the different retrievals. Therefore, we have rerun the retrievals for the open cell cases setting the drizzle to zero. The results are now shown in Fig. 8 c,d.

*2. The impact of shafts on retrievals is discussed, first in the algorithm description and then in the result discussions. But it is still not clear to me, especially in the discussions surrounding Tables 2 and 3, what part of the conclusions concern impacts from retrieval limitations or from actual microphysics differences during shafts. Please clarify the exact (expected) impact of shafts on retrievals, so that the readers can more clearly understand your results.*

Although the retrievals were performed on a time resolution of 1 minute, the results were analyzed statistically, in terms of shaft averages. This because instantaneous properties of drizzle shafts may be dominated by turbulent processes, however average properties are important to understand physical processes that affect the larger scales.
It is our opinion that the different characteristics between open cell and close cell systems evidenced in Tables 2-5 (2-3 in the previous version) are actual micro- and macro-physical differences and not artifacts of the retrievals. Tables 3 and 5 report results from the active part of the retrieval which is a fairly well-established technique. As for the passive retrievals (Tables 2 and 4) there are two main limitations that affect the results: the first limitation concerns the lack of sensitivity of the microwave to drop sizes smaller than ~ 100 μm. This limitation affects both open and closed call cases, however given that the frequency of occurrence of small drops is higher in closed cell systems it will probably lead to a larger underestimation of in-cloud DWP in these systems. The second limitation concerns the inability of the microwave to retrieve during the time of more intense precipitation. This will only affect the open cell cases and will result in an underestimation of the average shaft CWP and DWP. The quantification of the impact will likely require an LES model. We added these comments in section 4, lines 341-349.

---

## Author Response (AR2)

Colours/Figures: Generally, it is advised to choose colourblind-friendly colour schemes, as stated on the AMT website (https://www.atmospheric-measurement-techniques.net/for_authors/manuscript_preparation.html) under "Manuscript composition"/"Figure composition":
"7 - For maps and charts, please keep colour blindness in mind and avoid the parallel usage of green and red. For a list of colour scales that are illegible to a significant number of readers, please visit ColorBrewer 2.0."

*We changed the color scheme for Figures 1, 2, 5, 7, and 11 to comply with the journal guidelines.*

The rainbow colour scheme (used in Figures 1, 2, and 11) is unfortunately not a colourblind-friendly colour scheme (among other shortcomings of this colour table, see e.g. the open letter to the scientific community here: https://www.climate-lab-book.ac.uk/2014/end-of-the-rainbow/).

*Figures 1 and 2 were changed and now use the Viridis palette which is a suggested color scheme. Figure 11 was changed with a linear red color palette.*

In Figure 7 you use red and green in parallel, which should be avoided.

*Figure 7 and 5 were changed to use blue and purple instead of green.*

Please revise your figures accordingly!

Minor points:
p3 l113: PWV- please write out the abbreviation at the first occurrence. - *Done*

p10 l344: "closed call" -> typo cell. - *Done*

Figure 8 and 11: Please enlarge the numbers on x- and y-axes, and the axis annotations.
*The fonts of axis and annotations were increased of 2 points*